# Cytotoxic, Elastic-Plastic and Viscoelastic Behavior of Aged, Modern Resin-Based Dental Composites

**DOI:** 10.3390/bioengineering10020235

**Published:** 2023-02-10

**Authors:** Nicoleta Ilie

**Affiliations:** Department of Conservative Dentistry and Periodontology, University Hospital, Ludwig-Maximilians-University, D-80336 Munich, Germany; nilie@dent.med.uni-muenchen.de

**Keywords:** composites, cytotoxicity, aging, flexural properties, hardness, dynamic mechanical analysis

## Abstract

The development of resin-based composites (RBCs) is a delicate balance of antagonistic properties with direct clinical implications. The clear trend toward reducing filler size in modern RBCs solves esthetic deficiencies but reduces mechanical properties due to lower filler content and increases susceptibility to degradation due to larger filler–matrix interface. We evaluated a range of nano- and nano-hybrid RBCs, along with materials attempting to address shrinkage stress issues by implementing an Ormocer matrix or pre-polymerized fillers, and materials aiming to provide caries-protective benefit by incorporating bioactive fillers. The cytotoxic response of human gingival fibroblast (HGF) cells after exposure to the RBC eluates, which were collected for up to six months, was analyzed using a WST-1 assay. The microstructural features were characterized using a scanning electron microscopy and were related to the macroscopic and microscopic mechanical behaviors. The elastic-plastic and viscoelastic material behaviors were evaluated at the macroscopic and microscopic levels. The data were supplemented with fractography, Weibull analysis, and aging behavioral analysis. The results indicate that all RBCs are non-cytotoxic at adequate exposure. The amount of inorganic filler affects the elastic modulus, while only to a limited extent the flexural strength, and is well below the theoretical estimates. The nanoparticles and the agglomeration of nanoparticles in the RBCs help generate good mechanical properties and excellent reliability, but they are more prone to deterioration with aging. The pre-polymerized fillers lower the initial mechanical properties but are less sensitive to aging. Only the Ormocer retains its damping ability after aging. The strength and modulus of elasticity on the one hand and the damping capacity on the other are mutually exclusive and indicate the direction in which the RBCs should be further developed.

## 1. Introduction

The development of modern dental resin-based composites (RBCs) for direct restorations necessitates a complex balance between providing sufficient mechanical stability [1,2] and withstanding degradation [3] under the harsh conditions of the oral environment [4], in addition to adequate clinical functionality [5], biocompatibility [6], and esthetics [7].

The challenge in creating an RBC material is that most of the traits enumerated above are antagonistic. A fairly compelling example of this claim is the improved esthetics of RBC restorations over the last few decades, which has resulted in improved polishability through a reduction in filler size, but with the undesirable consequence that mechanical properties have deteriorated [2]. The antagonism is evident in this context, as the smaller the fillers, the better the polish and gloss, but reducing the filler size implicitly increases the surface area-to-volume ratio, thereby limiting the amount of filler that can be incorporated into the material. In addition, the larger surface area-to-volume ratio associated with small fillers also increases water uptake and consequent degradation of the filler/matrix interface [8], thereby compromising long-term mechanical properties [4]. Unfortunately, these initially theoretical and empirical observations [9] are directly reflected clinically since the causes of failure of RBC restorations in recent years indicate an increase in material and tooth fractures, whereas, in earlier RBC restorations, secondary caries was predominant [1].

One of the greatest disadvantages of RBCs is that they shrink during polymerization and, thus, generate stress on the tooth structure, which is clinically held as being responsible for the failure of the bond between the restorative material and the tooth [10]. This fact has prompted continued efforts to understand the complex relationship between polymerization shrinkage, material properties, restoration geometry, and the resulting tooth fracture or interface detachment [11,12]. The most important approaches to reduce polymerization shrinkage so far have been related to increasing monomer molecular mass [13]; reducing cross linking [14], polymerization rate, and gel point [10]; altering the amount of filler, shape, or surface treatment; or using porous and pre-polymer fillers [10]. Significant changes in a monomer matrix related to a reduction in polymerization shrinkage, such as Siloranes [15], have unfortunately not found long-term commercial acceptance. In contrast, Ormocers (organically modified ceramics) developed more than 20 years ago are still part of modern and clinically successful RBCs [16]. They are based on urethane dimethacrylate (UDMA) modifications [13] used to create large matrix monomers with few crosslinks [14]. The inorganic–organic monomers are synthesized from multifunctional urethane and thioether (meth)acrylate alkoxysilanes as sol–gel precursors. Alkoxysilyl groups of silane enable the formation of an inorganic Si-O-Si network through hydrolysis and polycondensation reactions, while methacrylate groups are available for photochemical polymerization [17].

While fundamental changes in a monomer system [15] or polymerization mechanisms [18] are observed less frequently in the development of modern RBCs, the most comprehensive and striking changes are caused by the optimization of the filler system [19]. This not only reduces the monomer content in an RBC and, thus, the polymerization shrinkage, but it also fine-tunes a large number of parameters, including mechanical properties [9], wear [20], translucency, opalescence, radiopacity, intrinsic surface roughness, esthetics, and handling properties [7]. In this context, nanotechnology has made a major contribution to the advancement of dental materials, enabling the production of nanodimensional filler particles, which are incorporated into resins either individually or as nanoclusters. The definition of nano-hybrid RBCs is a matter of controversy, as any RBC containing a small amount of nano-filler can be defined as such. The same applies to the designation of nano-RBCs since, in addition to nanofillers, they also contain agglomerates of nanofillers, which, however, reach a micrometer size. Both the use of nanoparticles and nanoparticle agglomerates enable the production of RBC materials that can be excellently polished, resulting in highly esthetic restorations that retain their gloss [7]. At the same time, these materials achieve good mechanical properties [9,16] and clinical abrasion resistance [20], which has made them one of the most widely accepted material categories at the moment.

In addition to the miniaturization of filler systems, the chemically inert fillers that are prevalent in commercial materials [9] have been supplemented with bioactive ones that can release ions to positively influence the remineralization of the tooth structure [21]. Since the elution of components of a material is associated with its degradation, concern has been raised related to the long-term behavior of this material category.

The present study aimed to evaluate comparatively modern RBCs that are pursuing different strategies in the development of the filler and monomer matrixes that they contain. The selected materials encompass Ormocers, nano- and nano-hybrid RBCs, as well as RBCs with bioactive or pre-polymerized fillers. To allow for a direct comparison, all materials were selected in the same shade A2.

It was hypothesized that the type of RBC has no effect on its (a) cytotoxicity; (b) mechanical behavior evaluated at the macroscopic and microscopic (elastic-plastic and viscoelastic behavior) levels; and (c) aging behavior.

## 2. Materials and Methods

### 2.1. Materials

Seven conventional light-cured, resin-based composites (RBCs) were selected, which belonged to different material categories described as Ormocers (AF), Giomers (BLS), RBCs with nano and agglomerated nanoparticles (FSE, FSB, and FSD), RBCs with pre-polymerized fillers (BEG and BLS), and RBCs with compact fillers (GSO) (Table 1).

The shade A2 was selected for all materials. The exposure time was set at 20 seconds, according to the manufacturer’s instructions, except for FSB that was cured, as recommended, for 40 seconds. A violet-blue LED (Light Emitting Diode) LCU (Light Curing Unit) was used for polymerization (Bluephase® Style, Ivoclar Vivadent, Schaan, Liechtenstein) with an irradiance of 1405 mW/cm² and measured with a spectrophotometer (MARC, Managing an accurate resin curing system; Bluelight Analytics Inc., Halifax, Canada).

The RBCs were characterized for cellular toxicity to human gingival fibroblasts in eluates collected up to 6 months, which was complemented by an analysis of mechanical behavior at different scales. The mechanical behavior was evaluated at a macroscopic level, specifically in terms of flexural strength, flexural modulus, and beam deflection, and was recorded in a three-point bending test. The data were supplemented with a fracture mechanism and reliability analysis. In addition, an instrumented indentation test was performed at the microscopic level in a quasi-static analysis to assess elastic and plastic deformation under indentation and in a dynamic-mechanical analysis to determine viscoelastic material behavior.

### 2.2. Cytotoxicity: WST-1 Assay

Cylindrical RBC specimens (5 mm in diameter and 2 mm in thickness) were prepared in previously sterilized molds. To perform this, the unpolymerized RBC material was placed in the mold, pressed between two polyacetate strips, and cured from the top only to simulate a clinical application, using the LED LCU described above. The exposure time was 20 seconds, except for FSB, which was cured for 40 seconds. The RBC specimens were then incubated together with a cell culture medium in 15 mL conical tubes (Falcon, Becton, Dickinson and Company, Franklin Lakes, NJ, USA), according to the recommendation of ISO 10993-12, which indorses a ratio of 117.8 mm^2^ sample surface area/mL cell culture medium [6]. All RBC materials were tested in triplicate; each triplicate was tested in four individual measurements. The cell culture medium was a Dulbecco’s Modified Eagle’s high-glucose medium (Sigma-Aldrich Co., St. Louis, MO, USA) supplemented with a 10% fetal bovine serum (FBS) and 1% penicillin/streptomycin (PenStrepFa, Sigma-Aldrich Co., St. Louis, MO, USA). The media without the test specimens served as the control and were collected similarly to the eluates. The collected eluates were replaced with 3 mL of fresh culture medium for additional storage. The collected eluates were frozen at −20 °C prior to testing.

The potential cytotoxicity of the tested materials was investigated using a WST-1 (4-[3-(4-Iodophenyl)-2-(4-nitro-phenyl)-2H-5-tetrazolio]-1,3-benzene sulfonate) colorimetric cell proliferation assay (Sigma-Aldrich Co., St. Louis, MO, USA). The tetrazolium salts of this assay are cleaved to formazan by the cellular mitochondrial dehydrogenases. There is a direct correlation between the number of metabolically active cells and the amount of formazan dye produced. The absorbance of the dye is measured spectrophotometrically [22].

The test used human gingival fibroblasts (HGF-1, ATCC® CRL-2014™). The cells were cultured in the above described medium using sterile cell culture dishes (CellStar®, Greiner Bio-One International GmbH, Kremsmünster, Austria) with the nominal size of 100/20 mm and incubated in a humidified CO_2_ incubator (HERACELL 150i, Thermo Scientific, Waltham, MA, USA) at 37 °C, 5% CO_2_, and a 95% air atmosphere. The medium was changed 3 times a week, and the cells were observed using an inverted phase-contrast microscope (Axiovert 40 C, Carl Zeiss AG). The cells were rinsed with Dulbecco’s Phosphate Buffered Saline (Sigma-Aldrich Co., St. Louis, MO, USA) after reaching an 80% confluence, and they were detached with a trypsin-EDTA solution (0.25% trypsin, 0.53% mM EDTA). Cell counting was performed using a mixture of 10 μL of cell suspension and 10 μL of trypan blue solution (T8154, Sigma-Aldrich Co., St. Louis, MO, USA) placed in a counting chamber (Neubauer Improved Haemocytometer, Paul Marienfeld GmbH & Co. KG, Lauda-Königshofen, Germany). The cells between passages 9 and 10 were used for all assays.

The cells were then seeded into 96-well cell culture plates (Cat.-No.655 160, CellStar®, Greiner Bio-One International GmbH, Kremsmünster, Austria) in 100 μL of cell culture medium at a density of 5000 cells per well. Twenty-four hours after seeding, the cell culture medium in each well was replaced with the eluates and the control media collected for each period, and the cells were incubated for 24 h in the humidified CO_2_ incubator. Ten μL of cell proliferation reagent WST-1 was added to each well, and the plates were incubated for two hours in the CO_2_ incubator at 37 °C before reading the optical density (OD). The wells without any cells served as the blank control. The OD was measured according to the manufacturer’s standard protocol at 440 nm and a reference absorbance at 600 nm [22], using a scanning multiwell spectrophotometer (Varioskan LUX Multimode Microplate Reader, Thermo Scientific, Waltham, MA, USA) and the SKanIT RE for Varioskan (Ver.2.2, Thermo Scientific, Waltham, MA, USA) computer software. The absorbance at 600 nm was subtracted from that obtained at 440 nm to account for background variation of the plate. The viability of the cells was calculated as a percentage of cell viability compared to the negative control (untreated cells) using the following Equation (1):(1)$$\%\;Viability=100\times \left(A440-\left.A600\right)\right.\;treated/\left(A440-\left.A600\right)\;control\right.$$

A cellular viability of 100% was attributed to the HGFs grown in the control cell culture medium. All RBC materials were tested in triplicate for each elution period, while four individual measurements were made for each collected eluate. A 10% ethanol added to the cell culture medium served as the control to check the validity of the test. The results’ interpretation was made according to ISO 10993-5 [23].

### 2.3. Three-Point Bending Test

A total of 140 (*n* = 20) specimens (2 mm × 2 mm × 18 mm) were prepared in a white polyoxymethylene mold according to the recommendation of ISO 4049:2019 [24] for a 3-point bending test. Light exposure followed the protocol specified in standard, which included irradiation at the top and bottom of the samples, with three light exposures overlapping an irradiated section by no more than 1 mm of the light-guide diameter to prevent multiple polymerizations. Immediately after curing, the specimens were removed from the mold and were ground with a silicon carbide paper (P 1200 grit, Leco Corp. SS-200, St. Joseph, MI, USA) to eliminate interfering edges or bulges, and then they were immersed in distilled water at 37 °C for 24 h in a dark environment. The flexural strength (FS), flexural modulus (E), and beam deflection (ε) were determined in a 3-point bending test according to NIST No. 4877, while considering a span of 12 mm [25]. Therefore, the samples were loaded in a universal testing machine until fracture (Z 2.5 Zwick/Roell, Ulm, Germany) at a crosshead speed of 0.5 mm/min. The force in bending was measured as a function of beam deflection, while the slope of the linear part of this curve was used to calculate the flexural modulus.

### 2.4. Light and Scanning Electron Microscopy (SEM) Evaluation

All fractured surfaces of the specimens tested in the 3-point-bendig test were analyzed under a stereomicroscope (Stemi 508, Carl Zeiss AG, Oberkochen, Germany) to determine the fracture pattern and the origin and were imaged with a microscope extension camera (Axiocam 305 color, Carl Zeiss AG, Oberkochen, Germany). The fractures were found to originate from either the volume (below the surface) or the surface (edges and corners) defects.

The microstructure of all materials was analyzed using a scanning electron microscopy (SEM, Zeiss Supra 55 V P, Carl Zeiss AG, Oberkochen, Germany) on samples that were prepared similarly as above (n = 3) and wet processed using an automatic grinder (EXAKT 400CS Micro Grinding System, EXAKT Technologies Inc., Oklahoma City, OK, USA) with gradually finer silicon carbide abrasive papers (1200, 1500, 2000, and 2400 grit). Surface preparation was completed by polishing the surface with a 1 µm diamond spray (DP-Spray, STRUERS GmbH, Puch, Austria).

### 2.5. Instrumented Indentation Test (IIT): Quasi-Static Approach (ISO 14577 [26])

An additional 70 (*n* = 5) slabs were prepared and processed as described above and stored at 37 °C for 24 h in distilled water in a dark environmental. Half of the specimens were tested after 24 h, and the other half were stored for a further 3 months at 37 °C in artificial saliva in a dark environment. Each surface was wet processed and polished before testing, as described above. An instrumented indentation test was performed in a quasi-static mode to assess the elastic-plastic material behavior by means of an automated micro-indenter (FISCHERSCOPE® HM2000, Helmut Fischer, Sindelfingen, Germany) equipped with a Vickers diamond tip. Three indentations were employed in each specimen, with 15 indentations per material. The test involved simultaneously recording the indentation depth and the indentation force during the whole indentation cycle, which considered an increase in the indentation force within 20 s from 0.4 mN to 1000 mN, a holding time of five s at the maximum force, and a subsequent reduction in force within 20 s at a constant speed. The integral of the indentation force over depth outlines the total mechanical work W_total_ (=∫Fdh). This is partially consumed as the plastic deformation work W_plast_, while the rest is released as the work of elastic recovery W_elastic_. The indentation modulus (E_IT_) was calculated from the slope of the tangent of the indentation depth curve at the maximum force. Further parameters, such as hardness, were then calculated by evaluating the impression created during the indentation. The resistance to plastic deformation is described by the indentation hardness (H_IT_ = F_max_/A_c_) and its more familiar correspondent, the Vickers hardness (HV = 0.0945 × H_IT_). Therefore, the projected indenter contact area (A_c_) was determined from the force–indentation depth curve, while considering the indenter correction based on the Oliver and Pharr model and described in ISO 14577 [26] and a previous calibration with sapphire and quartz glass. To characterize both plastic and elastic deformation, the universal hardness (or Martens hardness = F/A_s_(h)) was calculated by dividing the test load by the surface area of the indentation under the applied test load (A_s_).

### 2.6. Instrumented Indentation Test (IIT): Dynamic Mechanical Analysis (DMA)

Using the parameters determined in the quasi-static test as a prerequisite, the instrumented indentation test was used for a dynamic mechanical analysis to determine the viscoelastic material behavior. The specimens described above (n = 5) were used for testing. They were superimposed onto a quasi-static force of 1000 mN, and a low-magnitude oscillating force with an amplitude of 5 nm was employed. The frequency range was 0.5–5 Hz, following an exponentially increasing series (0.5 Hz, 0.7 Hz, 0.9 Hz, 1.1 Hz, 1.4 Hz, 1.8 Hz, 2.3 Hz, 3.0 Hz, 3.9 Hz, and 5.0 Hz) and including the frequency range of chewing in humans (0.94–1.7 Hz). Six randomly chosen indentations were performed for each specimen, amounting to 30 individual indentations per RBC and immersion condition. For each indentation, ten repeated measurements were performed for each frequency. For each of the described frequencies, the force oscillation generates oscillations on the displacement signal with a phase angle δ. The resulting sinusoidal response signal was separated into a real part (E′ = the storage modulus) and an imaginary part (E″= the loss modulus). By definition, E′ is a measure of the elastic response of a material behavior, and E″ characterizes the viscous material behavior. As a measure of the material damping behavior, the quotient E″/E′, which is also described as the loss factor (tan δ), was calculated.

### 2.7. Statistical Analysis

All variables were normally distributed, allowing a parametric approach to be used. A multifactor analysis of variance was applied to compare the parameters of interest (flexural strength; flexural modulus; beam deflection; Martens, Vickers, and indentation hardness; elastic and total indentation work; creep; indentation depth; storage, loss, and indentation moduli; and loss factor). The results were compared using one-way or multi-way analysis of variance (ANOVA) and Tukey’s honestly significant difference (HSD) post hoc test with an alpha risk set at 5%. A multivariate analysis (general linear model) assessed the effect of the parameters RBC, frequency, and aging (elution time and storage duration). The partial eta-squared statistic reported the practical significance of each term based on the ratio of the variation attributed to the effect. Larger values of partial eta squared (η_P_^2^) indicate a higher amount of variation accounted for by a model. A simple contrast estimation was used to compare the aged versus unaged groups for each parameter.

The flexural strength data were additionally analyzed using Weibull statistics. A common empirical expression for the cumulative probability of failure P at applied stress σ is the Weibull model (Equation (2)) [27]:(2)$$P_{f}\left(\sigma_{C}\right)=1-\exp\left[-{\left(\frac{\sigma_{C}}{\sigma_{0}}\right)}^{m}\right]\;$$
where $\sigma_{C}\;$ is the measured strength; m is the Weibull modulus; and $\sigma_{0}$ is the characteristic strength, defined as the uniform stress at which the probability of failure is 0.63. The double logarithm of this expression results in the following: $\ln\ln\frac{1}{1-P}=m\ln\sigma_{c}-m\ln\sigma_{0}$ By plotting $\ln\ln\frac{1}{1-P}$ vs. ln$\;\sigma_{C}$, a straight line results with the upward gradient m, whereas the intersection at the *x*-axis gives the logarithm of the characteristic strength [27].

## 3. Results

### 3.1. Cytotoxicity (WST-1) Assay

The parameters “RBC” (*p* < 0.001, η_P_^2^ = 0.517), “elution time” (*p* < 0.001, η_P_^2^ = 0.445), and the interaction between “RBC and elution time” (*p* < 0.001, η_P_^2^ = 0.382) all exerted significant effects on the cell viability, where the effect of RBC was slightly higher (higher eta squared values η_P_^2^) compared to the effect of elution time.

Figure 1 shows the percent viability of the HGF-1 cells exposed to the eluates from different RBCs at different elution times when compared to the corresponding negative control (CR). The eluates from the AF and the GSO show the lowest cell viability (84.7% and 88.4%, *p* = 0.967) after 24 h elution, followed by the BEG, while the FSE, FSD, FSB, BLS, and CR groups are statistically similar (*p* = 0.261). After 48 h elution, all groups tested show statistically similar or slightly higher cell viability than the negative control CR. The same applies to the 10-day eluates, but there is no significant difference between the RBCs or in comparison to the negative control for the elution times of up to six months.

### 3.2. Three-Point Bending Test

One-way ANOVA differentiates the flexural modulus (E) data into four homogeneous groups, with the highest values observed in the GSO group, followed by the groups consisting of FSB, FSD, and FSE (*p* = 0.601), then the BEG and BLS groups (*p* = 0.916), and finally the groups of BEG and AF (*p* = 0.310). The data are presented later in comparison to the indentation modulus. In contrast, the highest flexural strength (FS) values (Figure 2) are observed in the groups consisting of FSB, FSD, and FSE (*p* = 0.873), followed by the GSO, FSD, and FSE groups (*p* = 0.150), then the BEG group, and finally the BLS and AF groups (*p* = 0.947). In terms of beam deflection (Figure 3), the highest and lowest values are measured for the BEG group and the GSO group, respectively, while all other materials with intermediate values are statistically similar (*p* = 0.118).

The three identified patterns of fracture are quantified in Figure 4 for each material. In total, the highest percentage of failures originates from volume defects (73.6%), while 26.4% of defects originates from the surface as either edge (18.5%) or corner (7.9%) defects.

The reliability of the materials was evaluated using the flexural strength data by means of a Weibull statistic (Figure 5a,b, Table 2). High R^2^ values are observed for all groups (R^2^ > 0.90), indicating a very good fit with the Weibull model. Significantly higher m-values are observed for the RBCs containing nano and nano-agglomerates as fillers. In comparison, the reliability is lower for the Ormocers and the materials with PPF.

### 3.3. Scanning Electron Microscopy (SEM) Evaluation

The structural appearance of the filler systems was visualized by using a scanning electron microscopy with the electron backscatter diffraction mode (Figure 6). The backscattering method allowed a distinction to be made between the different types of fillers within one material. This includes compact glass fillers, as can be clearly seen in AF and GSO; round agglomerates of nanoparticles with clafted surfaces, as in FSB, FSD, and FSE; and large pre-polymerized fillers (PPF) that are well distinguishable from the rest of the matrix, as in BEG and BLS.

### 3.4. Instrumented Indentation Test (IIT): Quasi-Static Approach

A one-way ANOVA differentiates the indentation modulus (Figure 7), similarly to the flexural modulus data, into four homogeneous groups in the descending sequence GSO < (FSD, FSE, FSB; *p* = 0.456) < BEG, AF; *p* = 0.456) < (BEG, BLS; *p* = 0.373). This is also reflected in a good correlation of the parameters measured at the macroscopic (E) and microscopic levels (E_IT_) (Pearson correlation coefficient 0.89).

The variation in the HM and HV values and the corresponding statistical evaluation are shown in Figure 8, with a very good correlation between the parameters (Pearson correlation coefficient 0.98).

The indentation work is presented as the elastic and total work (Figure 9), the latter being the sum of the elastic and plastic indentation work. The ranking of the materials is in reverse order as observed for the elastic modulus and hardness data, being lower for GSO and increasing toward BLS. The correlation between the two given parameters is very good (Pearson correlation coefficient 0.97).

### 3.5. Instrumented Indentation Test (IIT): Dynamic Mechanical Analysis (DMA)

(a) 24 h storage (baseline)

The analysis of the influence of material and frequency on the measured parameters using multifactorial analysis unanimously shows a significant influence (*p* < 0.001). The binary combination also significantly affects all parameters, except for the H_IT_. The effect of the *RBC* is very strong on the H_IT_ (η_P_² = 0.988) and storage modulus (η_P_² = 0.975), followed by the loss factor (η_P_² = 0.774) and loss modulus (η_P_² = 0.465). *Frequency* exerts a very strong effect on the loss modulus (η_P_² = 0.916) and loss factor (η_P_² = 0.875), a moderate effect on the storage modulus (η_P_² = 0.669), while the effect on the H_IT_ (η_P_² = 0.164) is small.

Indentation hardness is the most discriminating parameter, with significant differences in the descending order shown in Figure 10. The variation pattern of hardness, as can also be seen from the multivariate analysis presented above, shows a very small variation with frequency for all the materials examined. The storage modulus (Figure 11) decreases slightly with frequency, with higher frequency being associated with lower storage modulus value. There is a good agreement in material order for storage modulus and hardness; however, storage modulus, when compared to hardness, cannot detect any difference between the FSB, FSD, and FSE groups.

In contrast, both the loss modulus (Figure 12) and the loss factor (Figure 13) are strongly affected by frequency, showing a rapid drop off up to 1.4 Hz and a slower rate drop for higher frequencies. Among them, the loss factor is more discriminatory. The lowest loss factor across all frequencies is identified for the GSO group, while the BEG and AF groups perform similarly and show the highest values.

(b) Effect of aging

The influence of aging and frequency on the parameters determined in the DMA analysis was determined for each analyzed material in a multifactorial analysis. Accordingly, aging significantly (*p* < 0.001) affects all parameters analyzed in each material, and the effect strength (partial eta-squared values) on the measured parameters is summarized in Table 3. The effect of aging is greatest (higher η_P_^2^ values, as shown in Table 3) for the nano-RBCs involving FSB, FSD, and FSE, while it is lowest for BLS. 

When assessed across the study design without differentiation within the materials, the effect of aging on the storage modulus is greatest (η_P_^2^ = 0.869); followed by the H_IT_ (η_P_^2^ = 0.759) and the loss modulus (η_P_^2^ = 0.205), while the effect on the loss factor is not significant (*p* = 0.576).

The variation in the measured DMA parameters with frequency in the aged and unaged (24 h storage) samples is shown below (Figure 14a–d) using the material FSD as an example.

Table 4 summarizes the simple contrast estimation (comparison of the aged versus unaged groups) for each parameter. The a priori contrasts show statistically significantly higher H_IT_ results in the aged groups compared to unaged groups for BEG (M_Diff_ = 25.6) and GSO (M_Diff_ = 13.6), while they are significantly lower in all others (negative M_Diff_ values). The storage modulus decreases with aging for the FSB, FSD, and FSE groups, while it increases slightly for all other materials. The loss modulus and loss factor also decrease with aging for all materials, except for AF.

## 4. Discussion

The chemical and microstructural design of an RBC is about balancing many antagonistic properties, with the consequence that each manufacturer focusing on different strategies when developing new products. These specific decisions and preferences can affect key material properties in different ways. Individual properties measured under laboratory conditions and the subsequent material behavior in a patient’s mouth are often only weakly, or not at all, connected, and the overall clinical success of a filling has proven to be multifactorial [2]. Therefore, the correct strategy to follow in the development of a material cannot be predefined. We selected some representative RBCs, in which either an increase in filler content to enhance mechanical properties, a modification of the polymer matrix to control shrinkage, the use of pre-polymerized, bioinert, and bioactive fillers, or the use of nanotechnology, were the focus, and we characterized them comparatively from various aspects. The selection included Ormocers (AF), Giomers (BLS), RBCs with nano and agglomerated nanoparticles (FSE, FSB, and FSD), RBCs with pre-polymerized (BEG and BLS), and RBCs highly filled with regular, compact fillers (GSO).

The hardness, flexural strength, and wear are among the few parameters that, albeit weakly, contribute to understanding the complex causality of a filling’s clinical behavior [2,5,28]. It should be noted that both historically and nowadays, these are the most frequently measured laboratory parameters, which is due in particular to the simplicity and standardization of their methods [29]. Studies correlating properties measured on artificially aged samples with clinical behavior are gaining importance due to the evidence that artificially aging protocols, such as thermal fatigue followed by immersion in alcohol solution, more accurately correlate with clinical behavior [5]. Likewise, the search for other in vitro material parameters that are able to predict clinical performance is of ongoing interest. In that regard, investigations, such as dynamic mechanical analysis, can add value by characterizing not only the elastic-plastic material behavior, but also the viscoelastic material behavior. In the characterization of dental materials, they have been used insufficiently to date, since the methodology and the corresponding devices are significantly more complex.

Since RBCs come into direct contact with biological tissues in a patient’s mouth, the assessment of their cytotoxicity and biocompatibility is of primary importance. RBC toxicity [30] can be induced by residual monomers [31] as a result of incomplete polymerization and impurities remained after resin synthesis, but also later degradation products associated with aging in the oral environment [12,32]. In addition, the oxygen-inhibited layer [33] can be a factor that can lead to an increase in toxicity since it is a layer in which radical polymerization has been partially inhibited by the reaction of initiator radicals with oxygen, possibly leaving an excess of unreacted monomers. It should be noted that the surface of cytotoxicity test samples is not ground, and, therefore, the oxygen-inhibited layer that forms on all surfaces of air-prepared RBC samples [33] is not removed. Moreover, to ensure clinical significance, the samples used in this study mimic the clinically common 2 mm thick increments and were exposed to light only from above, strictly following the light-curing conditions specified by the manufacturer (Table 1). With an exposure of 20 s (40 s for FSB) using an LCU with high irradiance, it can be assumed that the material has been sufficiently cured [34]. Besides, we selected the HGF-1 cell line for the test because it is widely used for the biological evaluation of dental materials, and the associated methodology is defined in detail in a corresponding standard (DIN EN ISO 10993-5 [23]). Further arguments in this regard are based on the fact that the HGF-1 cell line originates from gingival fibroblasts that are in direct contact with or very close to dental materials under clinical conditions [35]. Equally important, working with an established and well-characterized cell line provides a useful opportunity for a standardized and reproducible measurement of cell proliferation that is independent of individual donor differences. The WST-1 assay used in this study to assess cell viability features a broad linear range and yields a water-soluble cleavage product, which amount directly correlates with the number of metabolically active cells in the culture. While the test detects any type of cytotoxic effect of the material, its limitation is that it cannot pinpoint the exact cause of a potential toxicity. In addition, the WST-1 assay assesses the metabolic activity of the HGF cells but without differentiating whether the cells exhibit morphological changes, leaving scope for extending the investigations to other microbiological testing methods. The ISO 10993-5 [23] defines clear limit values for evaluating the cytotoxicity of materials and products. A substance is classified as cytotoxic if there is a reduction in cell viability of more than 30% compared to the negative control used in an experiment. Accordingly, the tested materials did not show significant cytotoxicity at any elution periods tested, as the lowest percent viability of the HGF-1 cells exposed to the eluates from the tested RBCs was 84.7% and 88.4% of the control, and these were recorded for AF and GSO after 24 h elution.

The most commonly observed strategy in the development of RBCs is to improve mechanical properties, generally along the lines of “the higher the better”. This desideratum can be achieved by increasing the proportion of inorganic fillers, which has an impact on the modulus of elasticity [9]. Packing a higher amount of filler into an RBC necessitates the use of larger fillers, in addition to using multimodal filler formulations to create densely packed structures [36]. For example, the amount of silica fillers required to achieve densely packed structures in RBCs has been theoretically estimated at 89% by weight [36], a value that corresponds to the amount of filler wt.% reported for GSO (Table 1). It should be noted that, in addition to silica, GSO also contains glass-ceramic fillers, which can alter the direct comparison to some extent. Therefore, estimates of the amount of filler by volume are of much greater importance. For low-dispersity particles, regardless of particle size, a maximum amount of 74.05% by volume has been estimated. The filler amount can be further improved by using a mixture of large and small particles or by using non-spherical particles [37]. It should be noted that all analyzed materials fall short of these estimates (Table 1), which indicates a potential for optimization.

Our comparative study clearly confirmed that increasing the proportion of inorganic fillers has a direct impact on the elastic modulus [9] at both the macroscopic and microscopic level. In this line, GSO, as the RBC with the highest inorganic filler content, achieved the highest modulus of elasticity, while the lowest values were measured in the materials with pre-polymerized filler and in the Ormocer-based RBCs. The presence of pre-polymerized fillers was confirmed by the SEM analysis (Figure 5) in both materials—BLS and BEG. The inorganic content of the fillers in these materials represents only part of the declared total filler content, thus being lower compared to the other analyzed materials. It represents 93.64% by weight of the total amount of filler (Table 1) for BEG, while the data for BLS are not available. In contrast, the low modulus of elasticity observed in the Ormocer-based RBCs may not be related to the amount of filler, but rather to the low crosslinks in the Ormocer matrix, which is due to the intention of reducing polymerization shrinkage stress [14]. It should be emphasized that not only is the correlation between the indentation modulus and the flexural modulus high, but the statistical differentiation of the materials is identical for both parameters. Even so, this observation should not be taken as being universally valid and sufficient to determine the modulus of elasticity in a single procedure, i.e., either macroscopically or microscopically, since effects, such as porosity, can have a greater impact at the macroscopic level than at the microscopic level, as defects at the microscopic level can be avoided by the targeted selection of the measuring points.

The increase in the proportion of inorganic fillers is not reflected equally in the measured mechanical properties. In contrast to the modulus of elasticity, the strength of the medium-loaded materials with nanofillers and nanofiller agglomerates approaches (FSD and FSE) or even exceeds (FSB) the flexural strength of GSO. In addition, the nano-RBCs proved to be significantly more reliable compared to GSO or to RBCs with pre-polymerized fillers. This behavior is well related to the microstructure as the size and nature of the nanofillers and nanofiller agglomerates enable a smooth surface with defects that are too small to initiate fracture. This was clearly confirmed by the fractography analysis, which identified volume located defects more frequently than surface defects as the initiators for fracture. In the same line, the literature provides a head-to-head comparison of nano-RBCs, the in the present study tested material GSO, and RBCs with pre-polymerized fillers in terms of abrasion and gloss, identifying the lowest wear rates and best gloss retention for the nano-RBCs, while the RBCs with pre-polymerized fillers performed the worst [38].

In contrast to the quasi-static parameters, the parameters characterizing the viscous material behavior, the loss modulus, and the loss factor, are higher in the Ormocer-based RBCs and the RBCs with pre-polymerized fillers. This behavior is well linked to the composition and microstructure of the materials since the main mechanism of stress dissipation is related to friction at the interphase boundary [39] and is also related to the polymer content. In this context, the lower crosslinks in the Ormocer matrix [14] act less constraining on the thermal movements of molecular chains [40], allowing for better stress dissipation. Similarly, the higher polymer amount in the RBCs with pre-polymerized fillers can lead to higher flexibility and a short adaptation time to the applied stress [40]. Confirming previous studies [16,41,42], the quasi-static parameters, such as strength, hardness, and modulus on the one hand and damping on the other, are mutually exclusive. Consequently, no material follows an ideal behavior, which should be characterized by high strength and high elastic modulus to resist deformation under load and, at the same time, high damping capacity to allow for stress dissipation.

Aging slightly but significantly degrades the mechanical properties, which is prevalently manifested in the materials with nanofillers and nanofiller agglomerates. This behavior is confirmed by previous studies [4] that demonstrate higher water uptake and greater degradation of the filler/matrix interface in RBCs with small fillers due to the degradation of the silanes that couple the inorganic fillers with the organic matrix [8]. For materials with nanofillers and nanofiller agglomerates, this degradation is manifested in a deterioration in H_IT_ (5.8 to 8.1%) and storage modulus (7.6 to 9.1%), as indicated by the a priori contrasts. Surprisingly, little to no degradation was observed in the materials with pre-polymerized fillers, since the boundary between the pre-polymerized fillers and the organic matrix is also defined as being prevalently hydrolytically degradable [43]. Hydrolytic degradation manifested in the materials with PPF as a slight (1.6%) reduction in H_IT_ for BLS only_._ On balance, for all the materials, except for AF, there is a decrease in the loss modulus and the loss factor with aging, indicating a slight deterioration in the ability to dissipate stress with aging. In particular, the loss factor decreases less for the materials with nanofillers and nanofiller agglomerates (2.4–4.8%) than for the materials with larger filler (12–13%), balancing the mentioned deterioration in H_IT_ and storage modulus. The Ormocer matrix proves to be more stable in this regard and retains its damping ability even after aging.

## 5. Conclusions

None of the materials tested is cytotoxic.

A high proportion of inorganic filler increases the modulus of elasticity, but only to a limited extent the flexural strength. The filler content is far lower than the theoretical estimates.

The RBCs with nanoparticles and agglomeration of nanoparticles show evidence of very good mechanical properties and excellent reliability, but experience a reduction in hardness and modulus of elasticity after aging, albeit slightly.

The pre-polymerized fillers reduce the mechanical properties but do not significantly affect the properties after aging.

The Ormocer matrix retains its damping ability even after aging.

The strength and modulus of elasticity on the one hand and the damping capacity on the other are mutually exclusive and indicate the direction in which the RBCs should be further developed.

## Figures and Tables

**Figure 1 bioengineering-10-00235-f001:**
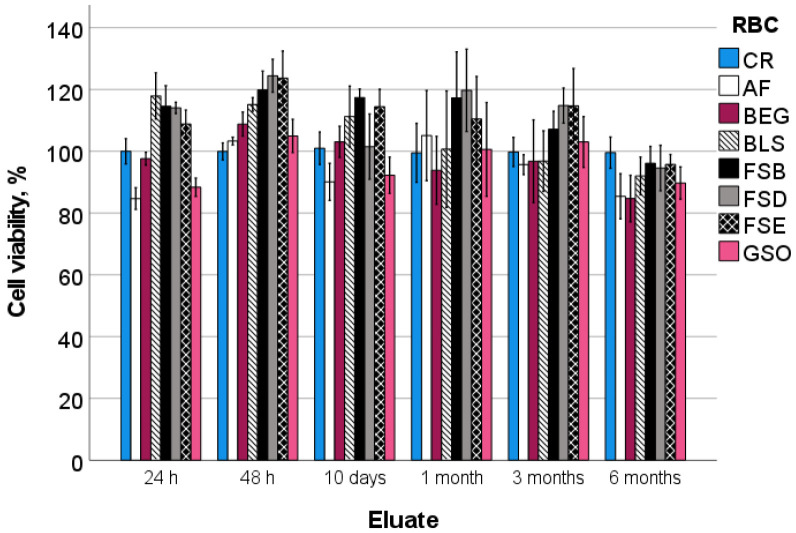
Cell viability (mean and standard deviation) in percentage of the negative control (CR), ordered by chronological elution times. Material abbreviation: AF = Admira Fusion; BEG = Brilliant EverGlow; BLS = Beautifil II LS; FSB = Filtek Supreme XTE Body; FSD = Filtek Supreme XTE Dentin; FSE = Filtek Supreme XTE Enamel; GSO = Grandio SO.

**Figure 2 bioengineering-10-00235-f002:**
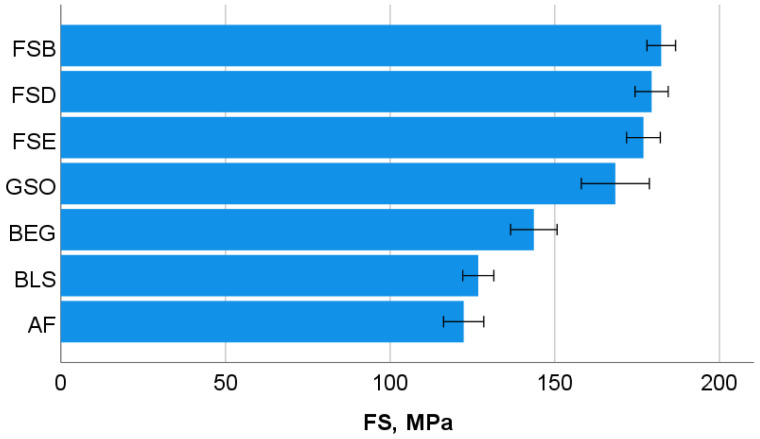
Three-point bending test: flexural strength (FS) with mean values and 95% confidence interval. Material abbreviation: AF = Admira Fusion; BEG = Brilliant EverGlow; BLS = Beautifil II LS; FSB = Filtek Supreme XTE Body; FSD = Filtek Supreme XTE Dentin; FSE = Filtek Supreme XTE Enamel; GSO = Grandio SO.

**Figure 3 bioengineering-10-00235-f003:**
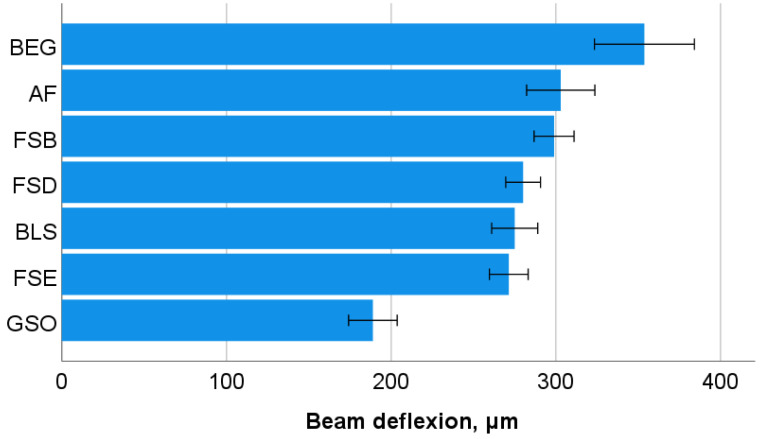
Three-point bending test: beam deflection (mean values with 95% confidence interval). Material abbreviation: AF = Admira Fusion; BEG = Brilliant EverGlow; BLS = Beautifil II LS; FSB = Filtek Supreme XTE Body; FSD = Filtek Supreme XTE Dentin; FSE = Filtek Supreme XTE Enamel; GSO = Grandio SO.

**Figure 4 bioengineering-10-00235-f004:**
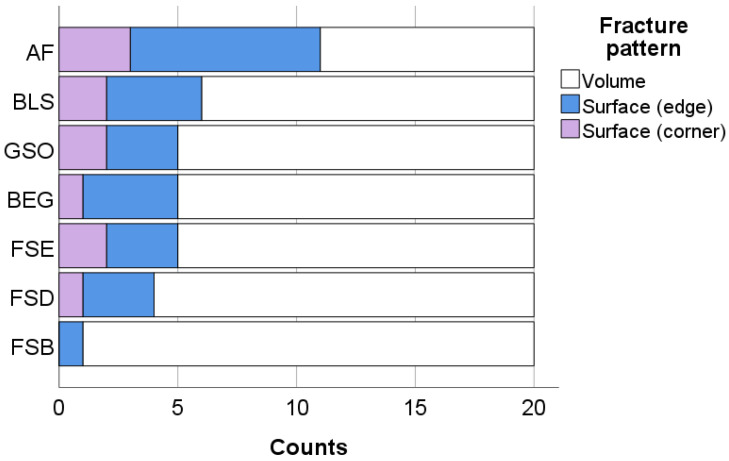
Fracture pattern as a function of material. Material abbreviation: AF = Admira Fusion; BEG = Brilliant EverGlow; BLS = Beautifil II LS; FSB = Filtek Supreme XTE Body; FSD = Filtek Supreme XTE Dentin; FSE = Filtek Supreme XTE Enamel; GSO = Grandio SO.

**Figure 5 bioengineering-10-00235-f005:**
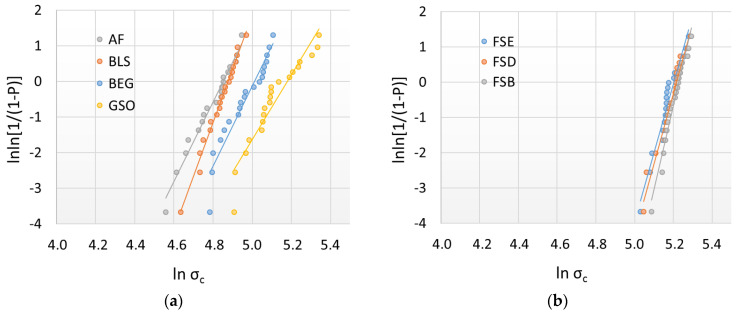
Weibull plot representing the empirical cumulative distribution function of the flexural strength data in the materials: (**a**) Admira Fusion; Beautifil LS; Brilliant EverGlow; and Grandio SO, and (**b**) Filtek Supreme XTE Enamel; Supreme XTE Dentin; and Supreme XTE Body.

**Figure 6 bioengineering-10-00235-f006:**
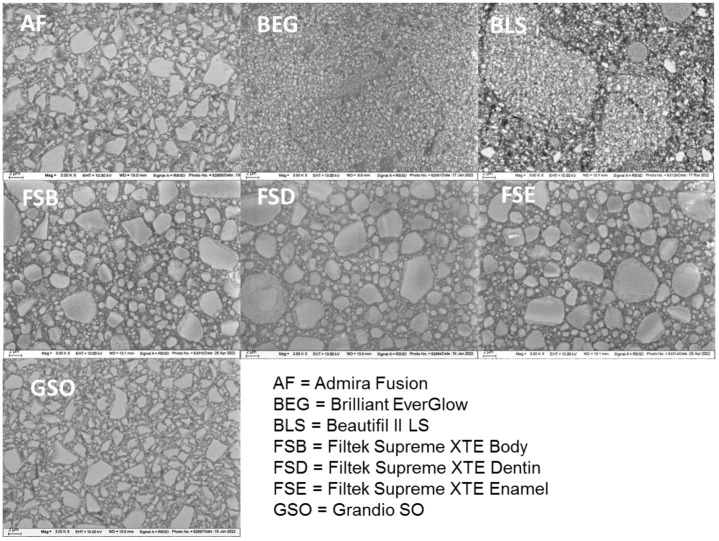
SEM images: structural appearance of the filler systems, as measured using the electron backscatter diffraction mode. Material abbreviation: AF = Admira Fusion; BEG = Brilliant EverGlow; BLS = Beautifil II LS; FSB = Filtek Supreme XTE Body; FSD = Filtek Supreme XTE Dentin; FSE = Filtek Supreme XTE Enamel; GSO = Grandio SO.

**Figure 7 bioengineering-10-00235-f007:**
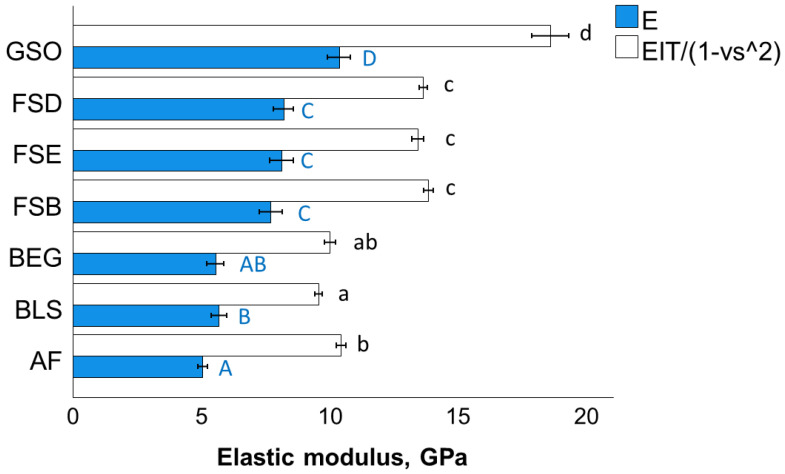
Indentation modulus (E_IT_) in comparison to the flexural modulus (E), as a function of RBC (mean values with 95% confidence interval; capital (E) and small letters (E_IT_) indicate statistically homogeneous subgroups; Tukey’s HSD test, α = 0.05). Material abbreviation: AF = Admira Fusion; BEG = Brilliant EverGlow; BLS = Beautifil II LS; FSB = Filtek Supreme XTE Body; FSD = Filtek Supreme XTE Dentin; FSE = Filtek Supreme XTE Enamel; GSO = Grandio SO.

**Figure 8 bioengineering-10-00235-f008:**
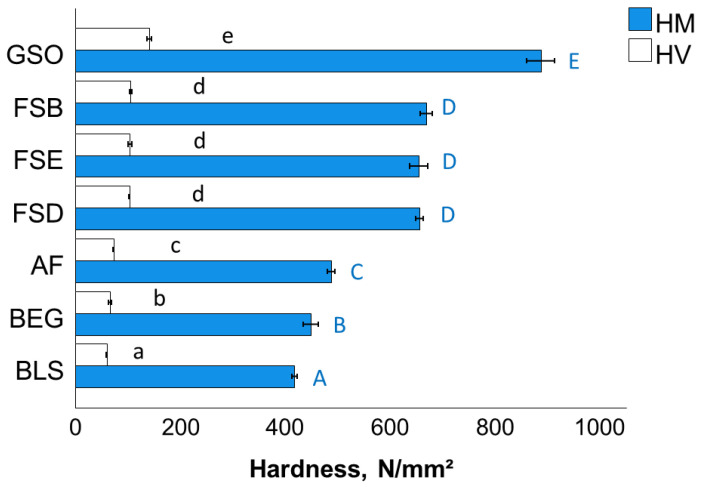
Martens (HM) and Vickers hardness (HV) as a function of RBC (mean values with 95% confidence interval; capital and small letters indicate statistically homogeneous subgroups; Tukey’s HSD test, α = 0.05). Material abbreviation: AF = Admira Fusion; BEG = Brilliant EverGlow; BLS = Beautifil II LS; FSB = Filtek Supreme XTE Body; FSD = Filtek Supreme XTE Dentin; FSE = Filtek Supreme XTE Enamel; GSO = Grandio SO.

**Figure 9 bioengineering-10-00235-f009:**
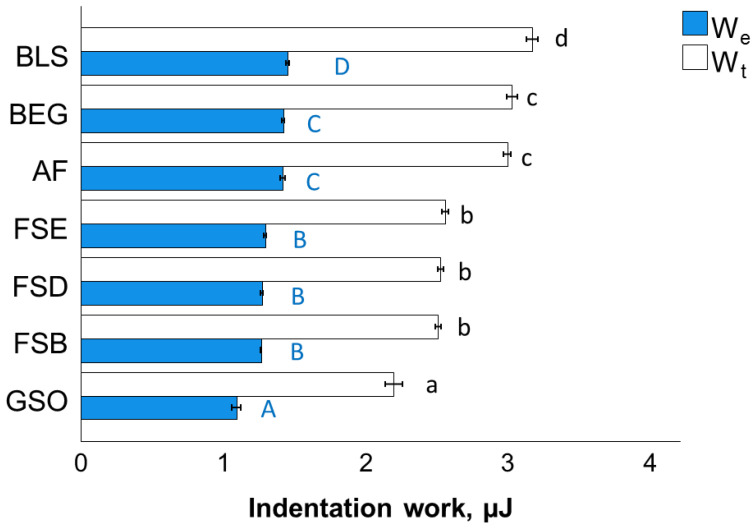
Elastic (W_e_) and total indentation work (W_t_) as a function of RBC and aging conditions (mean values with 95% confidence interval; capital and small letters indicate statistically homogeneous subgroups; Tukey’s HSD test, α = 0.05). Material abbreviation: AF = Admira Fusion; BEG = Brilliant EverGlow; BLS = Beautifil II LS; FSB = Filtek Supreme XTE Body; FSD = Filtek Supreme XTE Dentin; FSE = Filtek Supreme XTE Enamel; GSO = Grandio SO.

**Figure 10 bioengineering-10-00235-f010:**
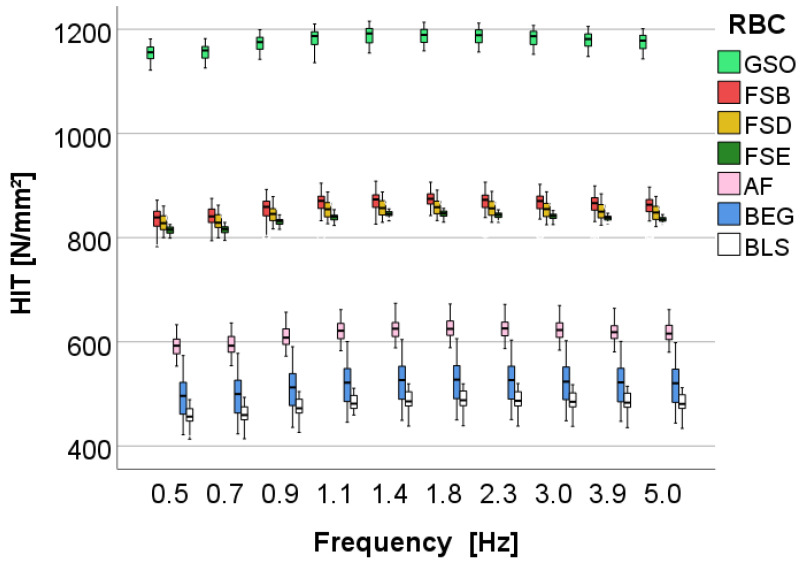
Dynamic mechanical analysis: indentation hardness (H_IT_) as a function of material and frequency. Material abbreviation: AF = Admira Fusion; BEG = Brilliant EverGlow; BLS = Beautifil II LS; FSB = Filtek Supreme XTE Body; FSD = Filtek Supreme XTE Dentin; FSE = Filtek Supreme XTE Enamel; GSO = Grandio SO.

**Figure 11 bioengineering-10-00235-f011:**
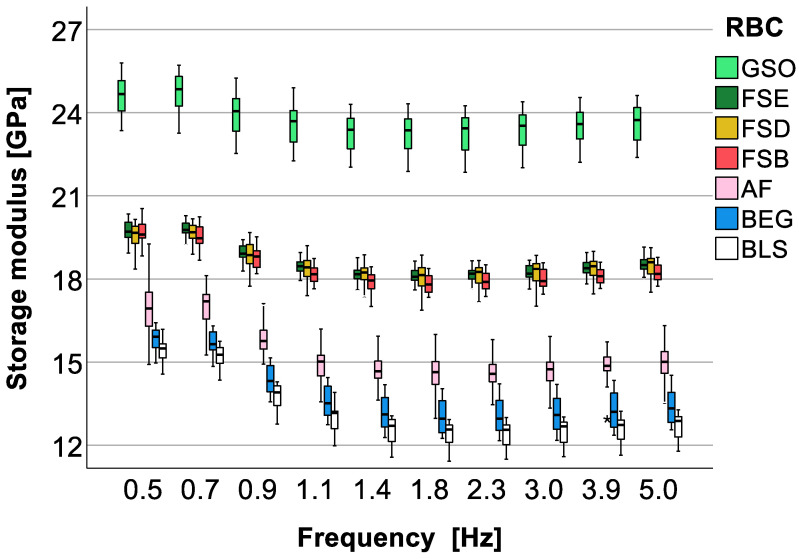
Dynamic mechanical analysis: storage modulus as a function of material and frequency. Material abbreviation: AF = Admira Fusion; BEG = Brilliant EverGlow; BLS = Beautifil II LS; FSB = Filtek Supreme XTE Body; FSD = Filtek Supreme XTE Dentin; FSE = Filtek Supreme XTE Enamel; GSO = Grandio SO (asterisks indicate outliers).

**Figure 12 bioengineering-10-00235-f012:**
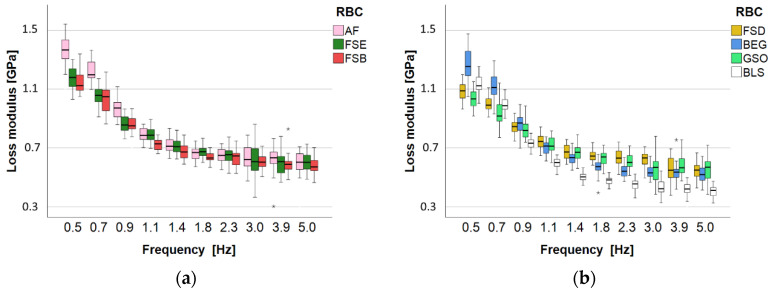
Dynamic mechanical analysis: loss modulus as a function of material and frequency for (**a**) AF, FSE, and FSB, and (**b**) FSD, BEG, GSO, and BLS. Material abbreviation: AF = Admira Fusion; BEG = Brilliant EverGlow; BLS = Beautifil II LS; FSB = Filtek Supreme XTE Body; FSD = Filtek Supreme XTE Dentin; FSE = Filtek Supreme XTE Enamel; GSO = Grandio SO (asterisks indicate outliers).

**Figure 13 bioengineering-10-00235-f013:**
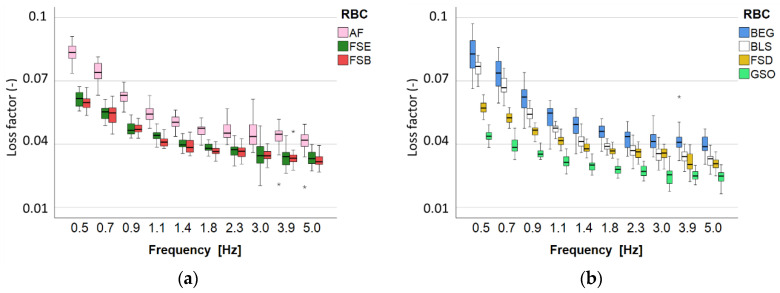
Dynamic mechanical analysis: loss factor as a function of material and frequency for (**a**) AF, FSE, and FSB, and (**b**) BEG, BLS, FSD, and GSO. Material abbreviation: AF = Admira Fusion; BEG = Brilliant EverGlow; BLS = Beautifil II LS; FSB = Filtek Supreme XTE Body; FSD = Filtek Supreme XTE Dentin; FSE = Filtek Supreme XTE Enamel; GSO = Grandio SO (asterisks indicate outliers).

**Figure 14 bioengineering-10-00235-f014:**
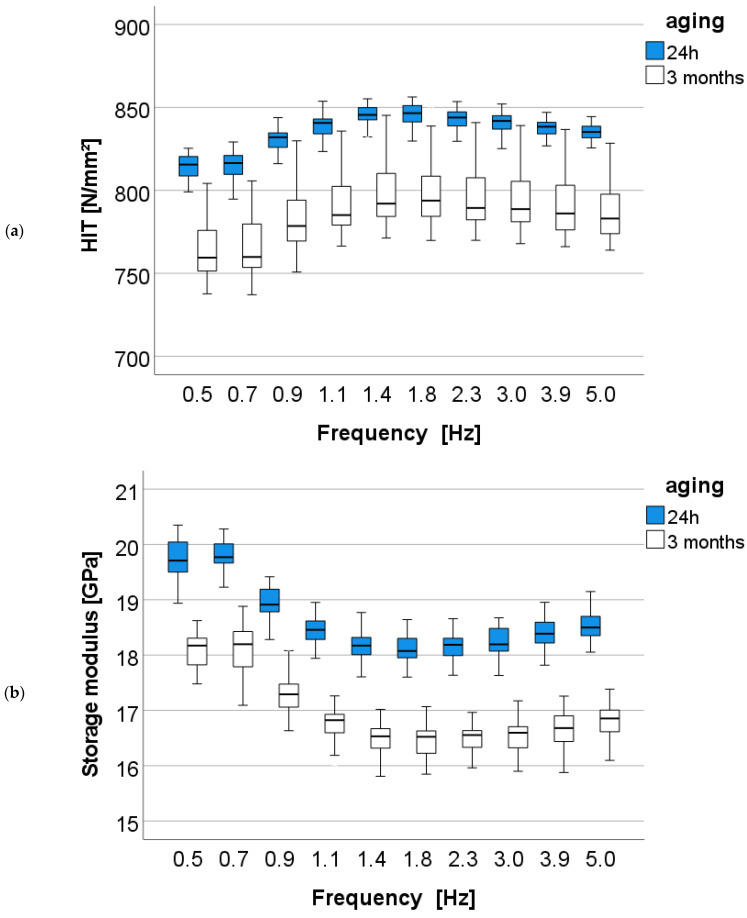
(**a**) Dynamic mechanical analysis: indentation hardness (H_IT_) as a function of aging and frequency exemplified for the material FSD (=Filtek Supreme XTE Dentin). (**b**) Storage modulus as a function of aging and frequency exemplified for the material FSD (=Filtek Supreme XTE Dentin). (**c**) Loss modulus as a function of aging and frequency exemplified for the material FSD (=Filtek Supreme XTE Dentin). (**d**) Loss factor as a function of aging and frequency exemplified for the material FSD (=Filtek Supreme XTE Dentin) (asterisks indicate outliers).

**Table 1 bioengineering-10-00235-t001:** Analyzed RBCs: Abbreviation (code), brand, manufacturer, shade, LOT, and composition, as indicated by the manufacturers.

Code	Material	Manufacturer	LOT	Monomer	Filler	
					Composition	wt./vol.%
AF	Admira Fusion	VOCO	1830088	Ormocer	SiO_2_	84/n.s.
BEG	Brilliant EverGlow	Coltene	I67156	TEGDMA, Bis-GMA	PPF, SiO_2,_ BaO-Al_2_O_3_-SiO_2_	79/64_t_ 74/56_i_
BLS	Beautifil II LS	Shofu	051813	UDMA, Bis-MPEPP; Bis-GMA, TEGDMA	PPF, S-PRG B_2_O_3_-F-Al_2_O_3_-SiO_2_	83/69
FSB	Filtek Supreme XTE Body	3M ESPE	N962746	Bis-GMA, Bis-EMA, UDMA, TEGDMA, PEGDMA	SiO_2_, ZrO_2_	78.5/63.3
FSD	Filtek Supreme XTEDentin	3M ESPE	N963106	Bis-GMA, Bis-EMA, UDMA, TEGDMA, PEGDMA	SiO_2_, ZrO_2_	78.5/63.3
FSE	Filtek Supreme XTE Enamel	3M ESPE	N963106	Bis-GMA, Bis-EMA, UDMA, TEGDMA, PEGDMA	SiO_2_, ZrO_2_	78.5/63.3
GSO	Grandio SO	VOCO	1828559	Bis-GMA, Bis-EMATEGDMA	Glass-ceramic, SiO_2_	89/73

Abbreviations: bis-GMA = bisphenol A glycol dimethacrylate; Bis-EMA = ethoxylated bisphenol A dimethacrylate; PEGDMA = poly (ethylene glycol) dimethacrylate; bis-MPEPP = bisphenol A polyethoxy methacrylate; TEGDMA = triethylene glycol dimethacrylate; UDMA = urethane dimethacrylate; PPF = pre-polymerized filler; SiO_2_ = silicon oxide (silica); ZrO_2_ = zirconium oxide; BaO-Al_2_O_3_-SiO_2_ = barium aluminosilicate glass; B_2_O_3_-F-Al_2_O_3_-SiO_2_ = boroaluminosilicate glass; S-PRG = surface pre-reacted glass ionomer filler; Ormocer = organically modified ceramic; “n.s.” not specified. Wt.% = percent by weight; vol.% = percent by volume; subscript t = total filler amount; subscript i = inorganic filler amount.

**Table 2 bioengineering-10-00235-t002:** Weibull parameter (m) and coefficient of determination of the regression model (R^2^). Linear regression was used to numerically assess the goodness-of-fit and estimate the parameters of the Weibull distribution.

RBC	AF	BLS	BEG	GSO	FSE	FSD	FSB
R²	0.98	0.93	0.99	0.90	0.95	0.96	0.96
m	11.1	11.2	15.4	9.1	19.4	19.8	23.8

**Table 3 bioengineering-10-00235-t003:** Effect strength of the factors *aging* and *frequency* on each individual RBC, defined by the parameter η_P_^2^. All effects are significant (*p* < 0.001).

RBC	Factor	H_IT_	Loss Factor	Storage Modulus	Loss Modulus
BEG	η_P_^2^ for aging	0.124	0.305	0.035	0.316
	η_P_^2^ for frequency	0.087	0.855	0.831	0.924
BLS	η_P_^2^ for aging	0.053	0.030	0.083	0.009
	η_P_^2^ for frequency	0.272	0.910	0.879	0.940
AF	η_P_^2^ for aging	0.464	0.060	0.008	0.082
	η_P_^2^ for frequency	0.342	0.893	0.717	0.927
GSO	η_P_^2^ for aging	0.087	0.222	0.552	0.098
	η_P_^2^ for frequency	0.202	0.765	0.457	0.797
FSB	η_P_^2^ for aging	0.739	0.039	0.840	0.244
	η_P_^2^ for frequency	0.236	0.823	0.810	0.863
FSD	η_P_^2^ for aging	0.843	0.027	0.792	0.104
	η_P_^2^ for frequency	0.327	0.879	0.678	0.897
FSE	η_P_^2^ for aging	0.759	0.001	0.869	0.205
	η_P_^2^ for frequency	0.403	0.837	0.781	0.870

Abbreviations: H_IT_ = indentation hardness; η_P_^2^ = partial eta squared. Material abbreviation: AF = Admira Fusion; BEG = Brilliant EverGlow; BLS = Beautifil II LS; FSB = Filtek Supreme XTE Body; FSD = Filtek Supreme XTE Dentin; FSE = Filtek Supreme XTE Enamel; GSO = Grandio SO.

**Table 4 bioengineering-10-00235-t004:** Contrast results (K-Matrix): comparison of aged vs. unaged groups for the measured DMA parameters. The contrast estimate (M_Diff_) is given for all significant comparisons (*p* < 0.05).

RBC		H_IT_	Loss Factor	Storage Modulus	Loss Modulus
BEG	Mean unaged	518.2	0.053	13.9	0.726
	M_Diff_	25.6	−0.007	0.2	−0.09
BLS	Mean unaged	478.6	0.046	13.2	0.614
	M_Diff_	−7.6	−0.002	0.2	−0.01
AF	Mean unaged	615.4	0.055	15.3	0.824
	M_Diff_	−30.8	0.002	0.1	0.05
GSO	Mean unaged	1180.1	0.031	23.7	0.710
	M_Diff_	13.6	−0.004	1.2	−0.05
FSB	Mean unaged	858.5	0.041	18.4	0.747
	M_Diff_	−69.8	−0.002	−1.4	−0.08
FSD	Mean unaged	848.6	0.041	18.5	0.734
	M_Diff_	−72.3	−0.001	−1.6	−0.04
FSE	Mean unaged	835.0	0.042	18.6	0.773
	M_Diff_	−48.2	n.s.	−1.7	−0.07

Abbreviations: RBC = resin-based composites with material cod as defined in Table 1; H_IT_ = indentation hardness; n.s. = not significant; M_Diff_ = contrast estimate. Material abbreviation: AF = Admira Fusion; BEG = Brilliant EverGlow; BLS = Beautifil II LS; FSB = Filtek Supreme XTE Body; FSD = Filtek Supreme XTE Dentin; FSE = Filtek Supreme XTE Enamel; GSO = Grandio SO.

## Data Availability

The datasets generated and/or analyzed during the current study are available from the corresponding author upon reasonable request.

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
