# Peer review of "Cytotoxic, Elastic-Plastic and Viscoelastic Behavior of Aged, Modern Resin-Based Dental Composites"

_bioengineering, 2023, doi:10.3390/bioengineering10020235_

Round 1

Reviewer 1 Report

This study assessed seven different modern resin-based composites (ormocers, nano, and nano-hybrid ones) all with the same shade A2. In these materials, the filler and the monomer matrixes they contain were evaluated. The manuscript contains seven keywords, three tables, fourteen figures, and forty-three references. Overall, it is a correct, complete, and well-conducted paper, although some remarks are made on different sections of the manuscript.

Keywords
The manuscript presents seven keywords. For keywords, where possible, please use Medical Subject Headings terms (MeSH Terms) and avoid the use of abbreviations. Strictly, only “aging” is a MeSH term. An alternative MeSH term proposed could be “composite resins” better than “resin-based composites”. What is the meaning of “DMA”? “dynamic mechanical analysis” or “dimethacrylate”. Please, clarify this. However, these suggestions about keywords are optional, not mandatory.

Other manuscript sections
Page 10, line 342. Please, consider replacing the abbreviation “Fig” with the complete word “Figure” as previously done to refer to the figures.
The last reference cited in the text is number 41, while two more, numbers 42 and 43 appear in the reference list. Please, place these new references in the text or remove them from the reference list.

References
Total number of the manuscript references: 43.
This is a complete and updated section. The reference format matches the journal’s reference format (ACS style guide). Please, check the discrepancy between the number of references cited in the text (41) and the number of references included in the list (43). 

Tables
Total number of the manuscript tables: 3.
In tables 2 and 3, please consider including a footer explaining the abbreviations as in table 1.

Figures
Total number of the manuscript figures: 14.
The figures have appropriate figure legends. Nevertheless, please consider explaining the abbreviations in each figure.

Author Response

All comments to the corresponding author have been addressed independently below. The authors’ rebuttal is always in BLUE and where changes have been added to the revised manuscript in light of the reviewer comments these are presented in RED.

The author would firstly like to thank the reviewers’ for taking the time to read and critically appraise the manuscript and secondly to thank the reviewers’ for their positive constructive comments in improving the work.

Comments and Suggestions for Authors

Reviewer 1 comments:

This study assessed seven different modern resin-based composites (ormocers, nano, and nano-hybrid ones) all with the same shade A2. In these materials, the filler and the monomer matrixes they contain were evaluated. The manuscript contains seven keywords, three tables, fourteen figures, and forty-three references. Overall, it is a correct, complete, and well-conducted paper, although some remarks are made on different sections of the manuscript.

Author’s response:  Thank you for your comments and appreciation.

Keywords
The manuscript presents seven keywords. For keywords, where possible, please use Medical Subject Headings terms (MeSH Terms) and avoid the use of abbreviations. Strictly, only “aging” is a MeSH term. An alternative MeSH term proposed could be “composite resins” better than “resin-based composites”. What is the meaning of “DMA”? “dynamic mechanical analysis” or “dimethacrylate”. Please, clarify this. However, these suggestions about keywords are optional, not mandatory.

Author’s response: Thank you for this hint! Keywords were adapted to the MeSH terms and the abbreviation DMA was written out. Although Dynamic Mechanical Analysis was not found in the MeSH database, it was used because it is an important keyword to describe the topic of the paper.

Other manuscript sections
Page 10, line 342. Please, consider replacing the abbreviation “Fig” with the complete word “Figure” as previously done to refer to the figures. The last reference cited in the text is number 41, while two more, numbers 42 and 43 appear in the reference list. Please, place these new references in the text or remove them from the reference list.

Author’s response: Thank you for pointing out the mistake.  Fig has been replaced with Figure and the reference list has been updated.

References
Total number of the manuscript references: 43.
This is a complete and updated section. The reference format matches the journal’s reference format (ACS style guide). Please, check the discrepancy between the number of references cited in the text (41) and the number of references included in the list (43). 

Author’s response: Thank you for pointing out the errors; the reference list has been updated.

Tables
Total number of the manuscript tables: 3.
In tables 2 and 3, please consider including a footer explaining the abbreviations as in table 1.

Author’s response: A footer with the used abbreviation has been added to Tables 2 and 3

Figures
Total number of the manuscript figures: 14.
The figures have appropriate figure legends. Nevertheless, please consider explaining the abbreviations in each figure.

Author’s response:  Thank you for pointing this out; the abbreviated terms were checked and supplemented if explanations were missing. Please consider modifications in the revised manuscript.

Reviewer 2 Report

The study is very interesting, well conducted and of great clinical interest. However, I add small suggestions for correction and/or changes.

Introduction

line 91 - I suggest authors read and reference the article: Mechanical and Tribological Characterization of a Bioactive Composite Resin . Appl. Sci. 2021, 11(17), 8256; https://doi.org/10.3390/app11178256

Authors should consider the null hypothesis. Not a positive hypotesis.

Materials and Methods

“…Seven conventional light-cured…” - Why did you choose 7 specimens for each group? What was the criteria?

Why did the authors not perform cell viability tests on odontoblasts, e.g. cell line (MDPC-23).

Results

Fig 6 - The figure legend must incorporate the description of the abbreviations

Discussion

Line 495 - I suggest authors read and reference the article: https://doi.org/10.1016/j.matdes.2014.04.056;

“…The WST-1 assay used in the study to assess cell viability…” - This test directly measures the metabolic activity of cells. This allows conclusions to be drawn about cell proliferation and viability. However, it is not a direct test of feasibility. Others would be more suitable: flow cytometry or evaluation with trypan blue dye. Why not choose one of these?

Conclusions

Conclusions should be more objective and synthetic.

References

I suggest they should add these related articles:

Mechanical and Tribological Characterization of a Bioactive Composite Resin . Appl. Sci. 2021, 11(17), 8256; https://doi.org/10.3390/app11178256

https://doi.org/10.1016/j.matdes.2014.04.056;

Author Response

All comments to the corresponding author have been addressed independently below. The authors’ rebuttal is always in BLUE and where changes have been added to the revised manuscript in light of the reviewer comments these are presented in RED.

The author would firstly like to thank the reviewers’ for taking the time to read and critically appraise the manuscript and secondly to thank the reviewers’ for their positive constructive comments in improving the work.

Comments and Suggestions for Authors

Reviewer 2 comments:

The study is very interesting, well conducted and of great clinical interest. However, I add small suggestions for correction and/or changes.

Author’s response:  Thank you for your comments and appreciation.

Introduction 

line 91 - I suggest authors read and reference the article: Mechanical and Tribological Characterization of a Bioactive Composite Resin . Appl. Sci. 2021, 11(17), 8256; https://doi.org/10.3390/app11178256

Author’s response:  thank you for the indicated reference, which I read with interest. However, the analyzed materials in the referenced paper are not bioactive resin composites like those analyzed in the present study. One material was Filtek Supreme XTE, a regular composite with inert fillers, the other is a GIC, and the third was Activa™ Bio-active Restorative, a hybrid material whose bioactivity is more related to the polymer matrix. I fear that the work is not suitable for comparison with the present study design.

Authors should consider the null hypothesis. Not a positive hypotesis.

Author’s response:  Null hypothesis was reformulated for more clarity; Please consider modifications in the manuscript text.

Materials and Methods

“…Seven conventional light-cured…” - Why did you choose 7 specimens for each group? What was the criteria?

Author’s response:  there was no criterion related to a fixed number of materials tested. As explained in the introduction, a number of different materials were chosen that followed different strategies in terms of design and composition. The selection includes materials designed for low shrinkage up to bioactive ones. The variety was the criterion.

Why did the authors not perform cell viability tests on odontoblasts, e.g. cell line (MDPC-23).

Author’s response: While acknowledging that I could have used several other methods for cytotoxicity analysis, I chose to study cell viability on human gingival fibroblasts (HGF-1, ATCC® CRL-2014™) since the connective tissue comes in direct contact with the restorative material. In addition, I have chosen a cell type and test that is widely accepted and prevalent in dental research and is part of a standard for dental materials that is unanimously accepted in the research community.

Results

Fig 6 - The figure legend must incorporate the description of the abbreviations

Author’s response:  The figure legends have been supplemented with all the requested details.

Discussion

Line 495 - I suggest authors read and reference the article: https://doi.org/10.1016/j.matdes.2014.04.056;

Author’s response: This is indeed a very interesting and relevant paper which is now cited in the revised manuscript.  

“…The WST-1 assay used in the study to assess cell viability…” - This test directly measures the metabolic activity of cells. This allows conclusions to be drawn about cell proliferation and viability. However, it is not a direct test of feasibility. Others would be more suitable: flow cytometry or evaluation with trypan blue dye. Why not choose one of these?

Author’s response: As mentioned above, I am aware of the shortcomings of the used test that have been expressed also in the discussion. While I cannot state whether the cell morphology has changed, the method used is reliable for toxicity testing of dental materials.

Conclusions

Conclusions should be more objective and synthetic.

Author’s response:  The conclusions have been restructured for clarity.

References

I suggest they should add these related articles

Mechanical and Tribological Characterization of a Bioactive Composite Resin . Appl. Sci. 2021, 11(17), 8256; https://doi.org/10.3390/app11178256 

https://doi.org/10.1016/j.matdes.2014.04.056;

Author’s response: Thank you for your suggestions; I added the reference: https://doi.org/10.1016/j.matdes.2014.04.056;

Reviewer 3 Report

The paper by N. Ilie describes the behaviour of seven different composites, based on resin, for dental applications. The Author clearly presents the mechanical and cytotoxic features of each of the examined composites, in correct English and with a rigorous approach. The materials' cytotoxicity has been investigated through WST-1 assay, as recommended by ISO 10993-5. The references and graphic components of the work are appropriate. Just a minor suggestion, i.e. please highlight that the WST-1 assay only assesses the metabolic activity of HGF cells, without telling us if the cells suffer from morphological alterations. This limitation of the assay should be pointed out and guide the Author to further detailed biological evaluations on the most promising composites.

Author Response

All comments to the corresponding author have been addressed independently below. The author's rebuttal is always in BLUE and where changes have been added to the revised manuscript in light of the reviewer's comments these are presented in RED.

The author would firstly like to thank the reviewers for taking the time to read and critically appraise the manuscript and secondly to thank the reviewers for their positive constructive comments in improving the work.

Comments and Suggestions for Authors

Reviewer 3 comments:

The paper by N. Ilie describes the behaviour of seven different composites, based on resin, for dental applications. The Author clearly presents the mechanical and cytotoxic features of each of the examined composites, in correct English and with a rigorous approach. The materials' cytotoxicity has been investigated through WST-1 assay, as recommended by ISO 10993-5. The references and graphic components of the work are appropriate. Just a minor suggestion, i.e. please highlight that the WST-1 assay only assesses the metabolic activity of HGF cells, without telling us if the cells suffer from morphological alterations. This limitation of the assay should be pointed out and guide the Author to further detailed biological evaluations on the most promising composites.

Author’s response:  Thank you for your comments and appreciation. The suggestion to highlight the shortcomings of the WST-1 assay was added to the revised paper. Please consider modifications in the manuscript text. Thank you.